# Comparative Investigation of the Corrosion Behavior and Biocompatibility of the Different Chemical Conversion Coatings on the Magnesium Alloy Surfaces

**Lingjie Meng [1,†], Xuhui Liu [2,†], Li Liu [3,†], Qingxiang Hong [1], Yuxin Cheng [1], Fei Gao [4,*], Jie Chen [1], Qiuyang Zhang [1,*] and Changjiang Pan [1,*]**

1   Faculty of Mechanical and Material Engineering, Jiangsu Provincial Engineering Research Center for Biomaterials and Advanced Medical Devices, Huaiyin Institute of Technology, Huai'an 223003, China
2   The Affiliated Huai'an Hospital of Xuzhou Medical University, Huai'an 223003, China
3   Department of Cardiology, The First College of Clinical Medical Sciences, China Three Gorges University, Yichang 443002, China
4   Chengdu Neurotrans Medical Technology Co., Ltd., Chengdu 610219, China
*   Correspondence: phil.gao@neurotrans.com.cn (F.G.); qyzhang@hyit.edu.cn (Q.Z.); panchangjiang@hyit.edu.cn (C.P.)
†   These authors contributed equally to this work.

**Abstract:** Due to their good biodegradability and biocompatibility, magnesium alloys are widely favored as the potential candidate for the biodegradable cardiovascular stent. However, the rapid degradation and the limited biocompatibility in vivo remain the main bottlenecks that inhibit their clinical applications. The construction of the chemical conversion coating on the magnesium alloy surface represents one of the effective strategies to control the degradation rate and enhance the biocompatibility. In the present study, the different chemical conversion layers were prepared on the magnesium alloy surface by chemical conversion treatment, including sodium hydroxide (NaOH), hydrofluoric acid (HF), phosphoric acid ($H_3PO_4$) and phytic acid ($C_6H_{18}O_{24}P_6$) treatment, and the corrosion behaviors and biocompatibility of the chemical conversion layers were comparatively investigated in detail. The results showed that the different chemical treatments can produce the different conversion layers on the magnesium alloy surfaces with a variety of physicochemical characteristics, corrosion resistance and biocompatibility, and all treatments can enhance the corrosion resistance to varying degrees. The hydrophilicity and corrosion resistance of the sodium hydroxide-treated magnesium alloy were the best among all the materials. Although the hydrofluoric acid-treated magnesium alloy had produced a hydrophobic coating, the corrosion resistance still needed to be improved. Magnesium alloys treated by sodium hydroxide showed a selective promotion of albumin adsorption, while the other samples simultaneously promoted albumin and fibrinogen adsorption. For the blood compatibility, the hemolysis rates of all of the treated materials were reduced to below 5%. The samples treated by phytic acid had the smallest hemolysis rate, and the NaOH-treated magnesium alloy had the least amount of platelet adhesion and activation. An appropriate microenvironment for cell growth could be achieved by the chemical conversion treatment, according to the results of the endothelial cell adhesion and proliferation, and the NaOH-treated surface showed the best endothelial cell growth behaviors among all of the samples. In summary, the corrosion resistance and biocompatibility of the magnesium alloy were significantly improved by the sodium hydroxide treatment, and thus this treatment can be used as a pretreatment for the surface modification of the magnesium alloy in order to further enhance the biocompatibility when used as the cardiovascular implants.

**Keywords:** magnesium alloy; surface modification; corrosion resistance; blood compatibility; cytocompatibility

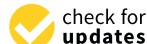



## 1. Introduction

Percutaneous coronary intervention (PCI) with the bare metal stents (BMS) or drug-eluting stents (DES) is the most popular way to treat stenotic coronary artery disease [1]. The insertion of the bare metal stents in the blood vessel can support and maintain the vessel lumen to maintain blood circulation [2]. The lower incidence of the intimal hyperplasia and in-stent restenosis is an advantage of DES over BMS [3]. Although conventional vascular stents made of non-degradable metals have the positive clinical outcomes and widespread applications, the stent will permanently remain in the body as foreign materials because they are non-degradable, which may cause a series of clinical side effects, such as the physical irritation over an extended period and the delayed endothelialization caused by chronic inflammation and neointimal atherosclerosis, etc. [4,5]. Therefore, the development of biodegradable stents in order to overcome the complications that exist with the permanent implants represents an important direction for the cardiovascular stents.

Magnesium and its alloys have attracted dominant interests as a novel and potential biodegradable metal for the cardiovascular stents because of its outstanding biocompatibility, biodegradability, and superior mechanical properties [6–10]. On the one hand, $Mg^{2+}$ level in human blood ranges from 0.70 to 1.11 mmol/L and it is the fourth prevalent cation in the human body. $Mg^{2+}$ could be absorbed by the body after the magnesium alloy implant degradation [4]. On the other hand, the chemical activity of the magnesium permits the in vivo degradation in the physiological environment, which can avoid secondary surgical removal and minimize the harm to the human body. Moreover, Mg also participates in protein synthesis and is essential to treat myocardial infarction and prevent neointimal atherosclerosis [11,12]. However, the high corrosion rate of magnesium and its alloy in vivo can cause a lot of adverse reactions, including excessive $Mg^{2+}$ release [13], local alkalinization, hydrogen production [14] and the enrichment of the secondary corrosion products [15], which may cause the premature loss of the mechanical properties of the implant before completing its physiological function, finally leading to the implantation failure.

It is well known that both the corrosion resistance and the biocompatibility of magnesium alloys are closely related to surface characteristics [16]. The major challenges for the magnesium alloys in cardiovascular implants are to control the in vivo corrosion rate [17,18] and to enhance their surface endothelial cell growth for endothelialization [19]. These issues must be resolved for magnesium alloys to be used safely in clinical applications. One of the most effective ways to control the degradation rate and regulate the bioactivity of the magnesium alloys is surface modification. Currently, the main strategies for improving the corrosion resistance of the magnesium alloy can be roughly divided into three aspects: mechanical modification (friction wear [20] and shot peening [21], etc.), physical modification (ion implantation [22], magnetron sputtering [23], iron plating [24] and laser melting [25], etc.), and chemical modification (surface chemical transformation [26], anodic oxidation [27], micro-arc oxidation (MAO) [28] and surface self-assembly [29], etc.). A protective layer of metal oxides or other compounds can be formed on the surface using a variety of surface modification techniques, but chemical conversion treatment represents the most effective way to produce the protective layer which can physically isolate the magnesium substrate from the corrosive medium. Chemical conversion coating is frequently employed in the field of the biomedical devices because of its easy-operation and low cost-effectiveness [30,31]. It can not only improve the adhesion of the final deposited coating to the substrate, but also enhance the corrosion resistance and the biocompatibility [32]. One study coated magnesium alloys with a variety of chemical conversion layers, such as $Mg(OH)_2$ [33], fluoride [34], molybdate [35], phosphate [36] and rare earth salts [37]. Gao et al. [38] reported that $Mg(OH)_2$ films had the properties of high hardness, superior corrosion resistance, and good biocompatibility, but the thin and loose $Mg(OH)_2$ conversion layer could not prevent the penetration of corrosive fluids. In vitro and in vivo testing by Mao et al. [34] showed that the fluoride treatment considerably enhanced the corrosion resistance and biocompatibility of Mg alloy stent materials. Fluoride conversion films, however, were less bioactive and less stable after the prolonged immersion. The toxicity of

molybdate conversion films has been reported to be unsuitable as the surface treatment for cardiovascular stent materials, despite their good corrosion resistance [35]. Although the phosphate coatings were insoluble in water and have outstanding biocompatibility, they were also very brittle and susceptible to breakdown [39]. The chemical stability and long-term insolubility of rare earth elements in physiological fluids were well established. The conversion film created by treating AZ31 magnesium alloy in cerium chloride solution was studied by Motemour et al. [40], and the results showed that it produced a thicker film with more coverage and superior corrosion resistance. Although a lot of the literature has reported that the chemical conversion layer can enhance the corrosion resistance and biocompatibility of the magnesium alloy, there is lack of the systematic comparative investigation between the different chemical treatments, especially when it is used for the cardiovascular implants.

AZ31B is a kind of widely used magnesium alloy for developing the cardiovascular implants. When the content of Al element in AZ31B magnesium alloy was about 3%, it could form sosoloid with magnesium, meanwhile Al could increase the strength and plasticity of magnesium alloy and improve the structure stability of the oxide film. Although Al is harmful to human body, the slow release of $Al^{3+}$ after the implantation will not cause great harm to the human body. The goal of this paper is to compare the physicochemical properties, corrosion resistance, and biocompatibility of the surface chemical conversion films created by four common chemical treatments, including sodium hydroxide (NaOH), hydrofluoric acid (HF), phosphoric acid ($H_3PO_4$) and phytate ($C_6H_{18}O_{24}P_6$), as well as to screen the coating preparation process for the best corrosion resistance and biocompatibility to meet the clinical biological application. To this end, the above-mentioned four chemical treatments were applied to produce the chemical conversion layers on the magnesium alloy, and the physiochemical properties, corrosion resistance and biocompatibility were comparatively investigated in detail.

## 2. Materials and Methods

### 2.1. Sample Preparation

An AZ31B magnesium alloy rod with the diameter of 12 mm was firstly cut into 5 mm slices, and then polished to a mirror finish with 400#, 600#, 1200#, 1500#, and 2000# sandpapers in turn. The samples were ultrasonically cleaned in acetone and ethanol for 10 min, respectively, and marked as Mg.

Alkali heat treatment: the cleaned Mg samples were immersed into a 3 M NaOH solution to treat 24 h in a water bath at 75 °C. The samples were cleaned by the deionized water and then dried and labelled as Mg-OH.

Fluoridation treatment: in order to clean and remove the surface impurities, Mg was first immersed into a solution of 50 g/L NaOH and 10 g/L $Na_3PO_4 \cdot 12H_2O$ for 15 min at room temperature. The samples were cleaned and then immersed in a 12 mol/L HF solution to treat 15 min. After being cleaned and dried, the samples were recorded as Mg-HF.

Phosphating treatment: the polished magnesium alloy was placed into a 3 M NaOH solution for 30 min to remove the surface impurities such as oil and organic substances, and then the sample was cleaned by the deionized water and dried. Finally, the sample was put into a mixed solution of 0.2 g/L $H_3PO_4$, 0.3 g/L $NaF_2$ and 0.5 g/L $BaH_2PO_4$ to treat 20 min at 90 °C. The sample was cleaned and dried, and the as-prepared sample was recorded as Mg-P.

Phytic acid treatment: the magnesium alloy was first washed by a 3 M NaOH solution for 30 min to remove the surface contaminants followed by cleaned with the deionized water. After dried, the sample was placed into 0.7 wt% phytic acid ($C_6H_{18}O_{24}P_6$) solution (PH 5) for treating 40 min in a 60 °C water bath. The sample was cleaned and dried. The sample was recorded as Mg-PA.

## 2.2. Surface Characterization

The surface atomic concentrations and elemental binding states of the modified magnesium alloys were analyzed by X-ray photoelectron spectroscopy (XPS, Quantum 2000; PHI Co., Chanhassen, MN). The surface microscopic morphology of the sample was observed by a scanning electron microscopy (SEM, FEI Quanta250, United States). The water contact angle was measured by a water contact angle meter (Krüss GmbH, Germany) to characterize the surface hydrophilicity/hydrophobicity; three parallel samples were measured and the average value was calculated.

## 2.3. Electrochemical Corrosion Behaviors

### 2.3.1. Potentiodynamic Polarization Curve

The potentiodynamic polarization curves of the samples were measured by a standard three-electrode system using a CHI660D electrochemical workstation (CHI Instruments, Inc., Shanghai, China). The test solution was Hank's simulated body fluid (SBF, compositions: NaCl, 8 g/L; KCl, 0.4 g/L; NaHCO$_3$, 0.35 g/L; CaCl$_2$, 0.14 g/L; Na$_2$HPO$_4$, 0.06 g/L; KH$_2$PO$_4$, 0.06 g/L; MgSO$_4$·7H$_2$O, 0.01 g/L; glucose, 1 g/L). The three-electrode system was composed of Ag/AgCl as the reference electrode, platinum wire as the auxiliary electrode, and the sample with an exposed area of 1 cm$^2$ as the working electrode. The sample was exposed to SBF solution for 10 min before the test to obtain a stable open circuit potential (OCP). The potentiodynamic polarization curve was measured at a scan rate of 1 mV/s and the corrosion potential (E$_{corr}$) and corrosion current density (i$_{corr}$) were obtained using the Tafel extrapolation method. The corrosion current density was used to calculate the annual corrosion depth according to the following formula [41]:

$$d = 3.28 \times 10^{-3}(M/n\rho)I_{corr} \qquad (1)$$

where d is the corrosion depth (mm/y), M is the gram atomic weight of Mg (24 g/mol), n is the atomic valence of Mg (n = 2), ρ is the density of Mg (1.74 g/cm$^3$), and I$_{corr}$ is the corrosion current density (μA/cm$^2$).

### 2.3.2. Immersion Experiment

To further evaluate the in vitro corrosion resistance of the blank and modified magnesium alloys, the samples were firstly sealed with the silicone rubber, exposing only 1 cm$^2$ area, and then the samples were immersed in 20 mL SBF solution and incubated at 37 ± 0.5 °C for 1, 3, 7 and 14 days, respectively. The SBF solution was changed every 2 days. After immersion, the samples were gently washed with the deionized water and dried in air. The surface corrosion morphologies and elemental compositions of the samples were observed by scanning electron microscopy (SEM, FEI Quanta250) and energy dispersive spectrometer (EDS) after the deposition of a gold layer, respectively.

### 2.3.3. pH Changes in Immersion Solution

The sealed samples with silicone rubber were immersed in 20 mL of pH 7.4 SBF solution for 1, 2, 3, 4, 5, 6 and 7 days, respectively, and the solution was changed every 2 days. The pH of the immersion solution was measured three times with a pH meter for each sample at the predetermined times and the average value was calculated. Finally, the pH change curve was plotted according to the values.

## 2.4. Protein Adsorption

The samples were firstly equilibrated with PBS solution for 2 h, then 2 mL of albumin and fibrinogen solution (1 mg/mL, 0.01 M PBS) were added, respectively. After fully adsorbed at 37 °C for 2.5 h, the samples were rinsed 3 times by PBS followed by adding 2 mL of sodium dodecyl sulfate solution (SDS, 1wt%) to ultrasonically desorb protein for 30 min. Finally, 100 μL of the solution was transferred into a 96-well plate, and the absorbance at 562 nm was measured. The protein adsorption amount was calculated

according to the standard curve, three parallel samples were measured, and the values were averaged.

### 2.5. Blood Compatibility

#### 2.5.1. Hemolysis Rate

The hemolysis rate test was performed according to the standard protocol of ISO 10993-4:2002. The healthy human whole blood containing 3.8 wt% sodium citrate was centrifuged 10 min at $1500 \times g$ rpm to obtain the red blood cells. The erythrocytes were diluted into 2% erythrocyte suspension with saline. The cleaned samples were placed into a 24-well plate, and then 2 mL of 2% erythrocyte suspension was added to each sample and incubated 3 h at 37 °C. Subsequently, 1 mL solution was centrifuged 5 min at 3000 rpm, and 200 μL of the supernatant was transferred into a new 96-well plate. The absorbance at 450 nm was measured by a microplate reader (Bio-Tek Eons). The 2% erythrocyte suspensions prepared with physiological saline and distilled water, respectively, were used as the negative and positive controls. The hemolysis rate was calculated as follows.

$$\text{Hemolysis (\%)} = (A - A_2)/(A_1 - A_2) \times 100\% \tag{2}$$

where A is the absorbance of the sample, $A_1$ is the absorbance of the positive control, $A_2$ is the absorbance of the negative control.

#### 2.5.2. Platelet Adhesion

Platelet-rich plasma (PRP) was first obtained by centrifuging the fresh whole blood from a healthy volunteer at $1500 \times g$ rpm for 10 min. PRP was covered on the sample surface and incubated 2.5 h at 37 °C. Then, the non-adhered platelets were washed off by phosphate buffer solution (PBS) and the adherent platelets were fixed 3 h with 2.5% glutaraldehyde solution at 4 °C. Finally, the samples were immersed in 50%, 70%, 90%, and 100% ethanol solutions in turn for 15 min each and dried at room temperature. The adhered platelets were observed by scanning electron microscopy (SEM, FEI, Quanta250, United States) after spraying a gold layer. At the same time, five randomly selected SEM images ($\times 3000$) for each sample were used for statistical counting of the platelets.

### 2.6. Endothelial Cell Behaviors

#### 2.6.1. Cell Adhesion

The samples sealed with silicone rubber were washed and placed into a 24-well cell culture plate for sterilizing 24 h under ultraviolet lamp. 0.5 mL of cell suspension ($5 \times 10^4$ cells/mL, ECV304, Cobioer, Nanjing, China) and 1.5 mL of cell culture medium were added and incubated at 37 °C with 5% $CO_2$ in an incubator for 1 and 2 d, respectively. The cells were first washed twice with saline, and then fixed 3 h with 2.5% glutaraldehyde at 4 °C. After washed again, the adhered cells were stained successively by 100 μL of rhodamine (10 μg/mL) for 20 min and 100 μL of 4,6-diamidino-2-phenylindole (DAPI, 500 ng/mL) for 5 min. After washed and dried, the fluorescent pictures of the adhered cells were taken by an inverted fluorescence microscopy (Carl Zeiss A2 inverted) in the dark.

#### 2.6.2. Cell Proliferation

The sealed samples were first placed in a 24-well plate and sterilized under UV light overnight. Then, 0.5 mL of $5 \times 10^4$ cells/mL endothelial cells and 1.5 mL of culture medium were added to each well and incubated for 1 and 2 d, respectively. Subsequently, the samples were transferred into a new 24-well plate, 0.5 mL of 10% CCK-8 solution was added to each sample and incubated 3.5 h at 37 °C. Finally, 200 μL of the solution was transferred into a 96-well plate, and the absorbance at 450 nm was measured with a microplate reader (Bio-Tek Eons). Three parallel samples were measured, and the values were averaged.

### 2.7. Statistical Analysis

All the data were expressed as mean ± standard deviation (SD) and statistically analyzed using SPSS 12.0. Statistically significant differences were determined by one-way analysis of variance (ANOVA), and $p < 0.05$ was considered to be statistically significant.

## 3. Results and Discussion

### 3.1. Surface Characterization

It is well known that the chemical treatment can produce the chemical conversion layers on the magnesium alloy surface concurrently with the introduction of new elements. In this study, the surface elemental concentrations and their valence states after the chemical treatments were firstly determined by X-ray photoelectron spectroscopy (XPS). Figure 1 and Table 1 show the survey spectra and elemental concentrations of the different samples, respectively. The high-resolution spectra of the different conversion layers are shown in Figure 2. The blank magnesium alloy was mainly made of the elements O, C, and Mg, indicating that a naturally occurring oxide layer can be produced due to the chemical activity of magnesium. The $Mg_{1s}$ spectra had a distinctive peak of $Mg^{2+}$ at 1305.91 eV. The occurrence of the C element (284.72 eV) can be attributed to the presence of hydrocarbons in air. The presence of O element (532.75 eV) should come from the hydrocarbon contaminations or MgO created by air oxidation [42]. As compared to the blank Mg, the O content on Mg-OH surface increased obviously while the content of C and Mg dropped substantially, suggesting that the sodium hydroxide treatment can produce a $Mg(OH)_2$ layer on the magnesium alloy surface. The existence of $Mg(OH)_2$ was also indicated by the $O_{1s}$ single peak at 531.33 eV on the survey spectra. The high-resolution $C_{1s}$ can be fitted into three peaks, the peak at 284.56 eV (C-C bond), the peaks at 285.25 and 287.91 eV, which can be attributed to the occurrence of $MgCO_3$. The formation of $MgCO_3$ may be the results of $CO_2$ in the air diffusing into the inner layer and reacting with the magnesium. The $Mg_{1s}$ spectra may be divided into two peaks: the peak at 1304.65 eV corresponding to $MgCO_3$ and the peak at 1307.44 eV belonging to $Mg(OH)_2$ [43]. For Mg-HF, the elements of C, F, O, and Mg were detected on the survey spectra, and $F_{1s}$ single peak at 690.79 eV were discovered in the high-resolution spectra, which corresponded to $MgF_2$, showing that $MgF_2$ was the primary component of the chemical conversion film. The $O_{1s}$ peak can be attributed to the presence of a minor number of hydroxides, and the structure could be made of a mixture of $Mg(OH)_2$ and $MgF_2$ or a partial replacement of $F^-$ in $MgF_2$ by $OH^-$ [44]. The $Mg_{1s}$ spectra can be divided into two peaks: one was the peak at 1309.27 eV corresponding to $Mg(OH)_2$, and the other peak at 1310.75 eV of $MgF_2$. The presence of new peaks of $P_{2p}$ and $Ba_{3d}$ on Mg-P surface showed that the oxide layer was successfully replaced by the phosphate coating. The $O_{1s}$ spectra can be divided into two peaks: the peak at 532.82 eV arising from $H_2O$ in the form of crystallization and the peak at 534.22 eV of P-OH group [45]. The high-resolution $P_{2p}$ spectra is divided into five peaks: the peak of $PO_4^{3-}$ at 133.32 eV, the peaks of $HPO_4^{2-}$ at 134.03 eV and 134.73 eV, and the peaks of $H_2PO_4^-$ at 135.91 eV and 136.45 eV, demonstrating that P element was mainly existed in the surface as $Mg(H_2PO_4)_2$. Four peaks were used to match the $Mg_{1s}$ spectra: MgO, $MgHPO_4$, $Mg_3(PO_4)_2$ and $Mg(H_2PO_4)_2$ at 1303.66 eV, 1304.25 eV, 1305.8 eV and 1306.83 eV, respectively [46]. As a result, $Mg_3(PO_4)_2$ and $Mg(H_2PO_4)_2$ were the main constitutes of the chemical conversion layer. The phytate-treated sample (Mg-PA) contained the elements O, C, P and Mg according to the survey spectra, and the presence of P (3.69%) demonstrated that the chemical conversion coating was successfully formed on the magnesium alloy surface. The $O_{1s}$ spectra can be fitted into two peaks. One was the peak at 530.02 eV belonging to $PO_4^{3-}$ and $HPO_4^{2-}$, proving that some of the O elements were from phytate. The other peak was the peak of magnesium oxides and hydroxides at 533.38 eV. The phytate radicals on oxides and $CO_2$ were represented by the fitted peaks at 284.77 and 288.74 eV on the $C_{1s}$ spectra, respectively. $P_{2p}$ displayed a single peak of $PO_4^{3-}$ at 128.89 eV.

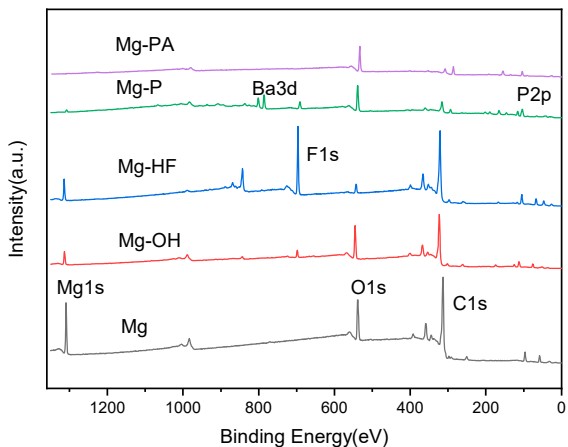

**Figure 1.** XPS survey spectra of the different samples.

**Table 1.** Surface atomic percentages of the different samples.

| Samples | Mg | C | O | P | F | Ba |
|---|---|---|---|---|---|---|
| Mg | 6.3 | 41.27 | 52.43 | - | - | - |
| Mg-OH | 2.99 | 29.99 | 67.02 | - | - | - |
| Mg-HF | 0.35 | 44.77 | 28.09 | - | 26.78 | - |
| Mg-P | 0.25 | 18.74 | 67.57 | 5.08 | 7.13 | 1.23 |
| Mg-PA | 0.24 | 43.89 | 52.18 | 3.69 | - | - |

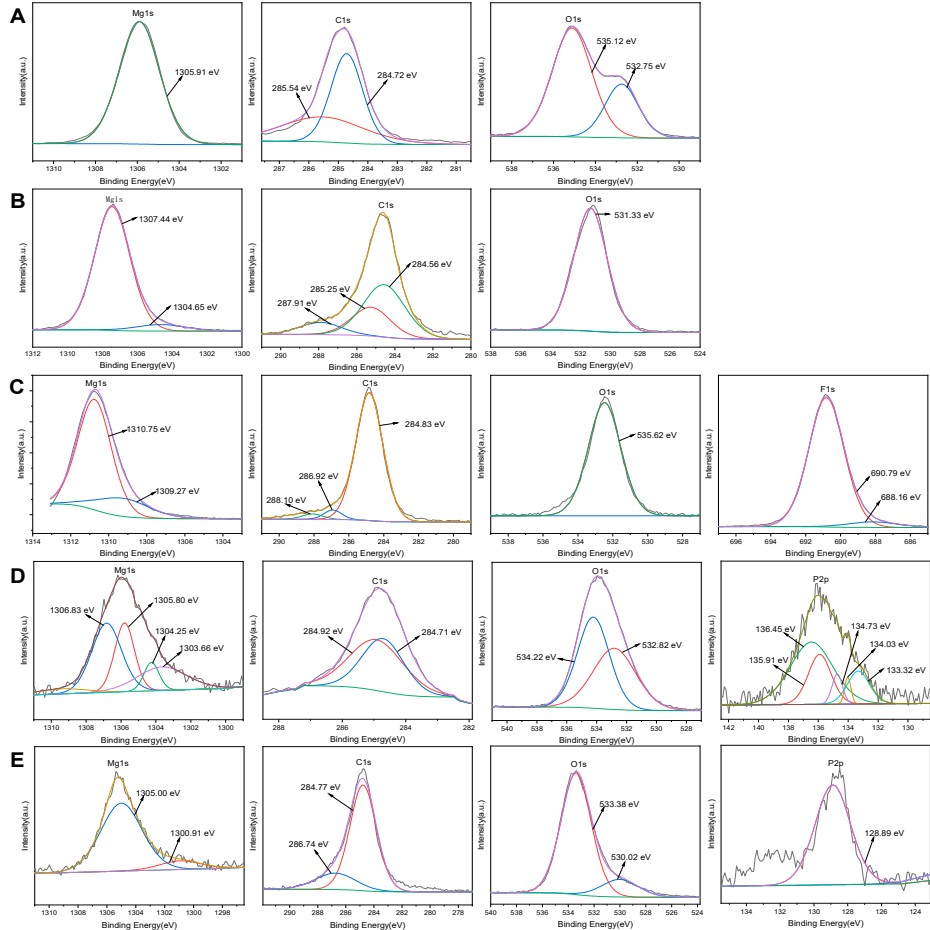

**Figure 2.** The high-resolution spectra of the main elements on the surfaces of different conversion films: (**A**) Mg; (**B**) Mg-OH; (**C**) Mg-HF; (**D**) Mg-P; (**E**) Mg-PA.

The surface morphologies of the different chemically treated magnesium alloys were observed by SEM, as illustrated in Figure 3. After polishing, the pristine magnesium alloy surface was relatively smooth. The surface of Mg-OH had a dense and homogenous $Mg(OH)_2$ layer, which slightly increased the surface roughness. The HF treatment can produce a dense and smooth coating on the surface, and there were also a few scattered irregular pores that may be due to the hydrogen generation and could be filled by $MgF_2$ and MgO particle precipitation [47]. A dense and lamellar-structured conversion film with visible cracks and pores can be observed on the Mg-P surface, which may result from the internal stresses during drying or when sodium fluoride is added, the entry of the F element results in the creation of hydrogen [48]. With the exception of a few small cracks, the surface chemical conversion film of the phytate-treated sample (Mg-PA) was intact and homogeneous, which was due to the fact that phytate contained twelve hydroxyls and six phosphate groups which can react with the dissolved metal ions of the magnesium alloy to form the water-insoluble magnesium phytate chelate. However, this reaction also produces a small amount of hydrogen, which passes through the surface layer to cause surface cracking [49]. In addition, it can be seen that both Mg-P and Mg-PA exhibited surface cracks that may be the result of substrate corrosion, and the presence of conversion layer flaws or poor crystallinity allows corrosion ions to diffuse into the surface of the substrate, resulting in the continuous anodic dissolution of the magnesium substrate [50]. The combination of XPS and SEM results demonstrated that the chemical conversion treatment can successfully create the different surface layers on the magnesium alloy surface. These coatings can isolate the substrate from the corrosive medium and stop reactive ions in solution from penetrating into the magnesium surface, which is helpful in reducing the corrosion rate of the magnesium substrate.

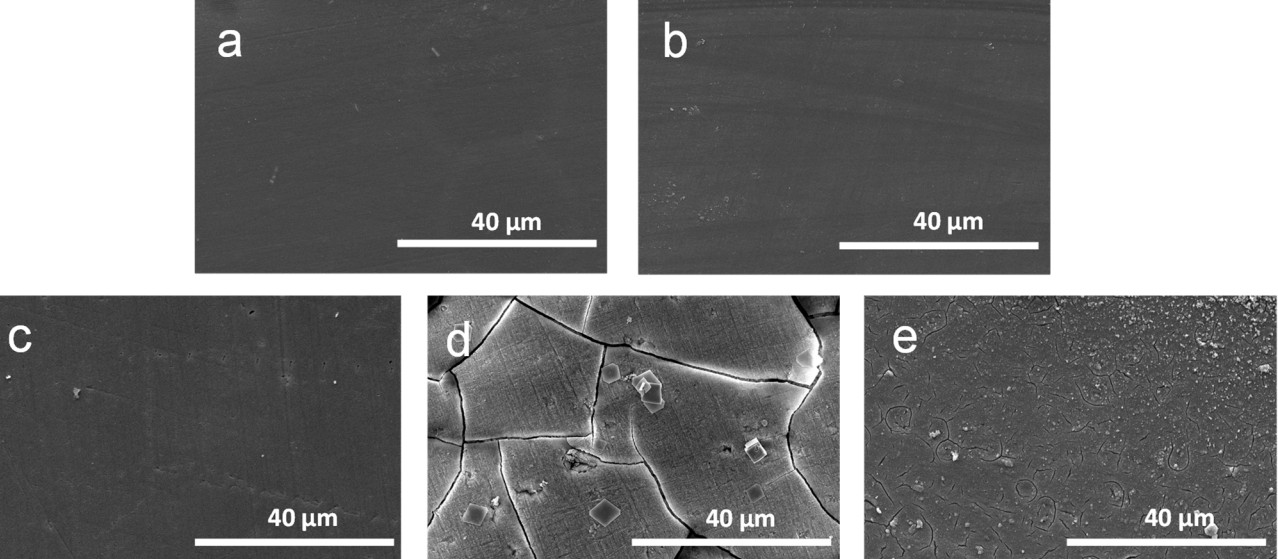

**Figure 3.** SEM images of the surface morphologies of the different samples: (**a**) Mg; (**b**) Mg-OH; (**c**) Mg-HF; (**d**) Mg-P; (**e**) Mg-PA.

Generally speaking, a good surface hydrophilicity is advantageous in improving the biocompatibility of the biomaterials because of the prolonged contact between the implant and the physiological environment in the body, as well as the substantial volume of water present in a human body. The surface hydrophilicity/hydrophobicity of bio-materials is often characterized by the water contact angle. In general, the hydrophilic surface is easy to wet when being in contact with water, and the water contact angle typically decreases as the surface roughness increases. Figure 4 displays the water contact angle results of the magnesium alloy treated by the different chemical treatments. As shown in Figure 4, the pristine magnesium alloy had a water contact angle of 56.3° and was

relatively hydrophobic. The alkali heat treatment can produce a lot of hydroxyl groups on the magnesium alloy surface, which can combine with the water molecules to form hydrogen bonds and endow with a better hydrophilicity, resulting in a hydrophilicity of 12.5° for the Mg-OH. It has been demonstrated that the hydrophilic surface is useful for lowering the platelet adhesion and promoting cell adhesion and tissue growth [51]. The hydrophobic compound $Mg(OH)_xF_{2-x}$ on the surface of Mg-HF can cause an increase in the water contact angle to 103.2°, indicating that HF treatment can create a hydrophobic surface. Additionally, the increase in hydrophobicity also results in less surface adhesion between water droplets, which also raises the water contact angle. The water contact angles of Mg-P and Mg-PA were 12.7° and 19.7°, respectively. It was concluded that the presence of hydrophilic P-OH groups in $Mg_3(PO_4)_2$ and the hydroxyl groups in $Mg(OH)_2$ layer as well as the increased surface roughness can significantly reduce the water contact angle. Both the XPS results and the SEM images supported this conclusion.

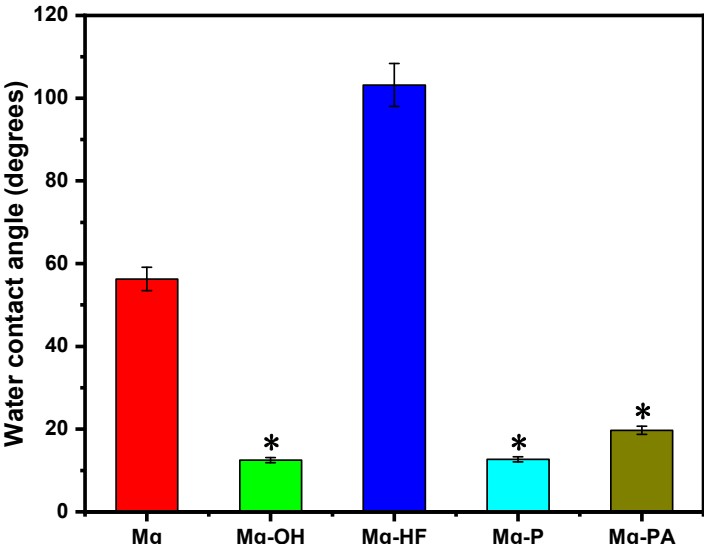

**Figure 4.** Water contact angles of the different modified magnesium alloy surfaces. The values are expressed as mean ± standard deviation (n = 3). The values of Mg-OH, Mg-P and Mg-PA are significantly lower ($p < 0.05$) than those of the pristine magnesium alloy in terms of hydrophilicity.

*3.2. Electrochemical Corrosion Behaviors*

The chemical structure, surface morphology, and microstructure of the surface chemical conversion layers are all strongly correlated with the corrosion resistance of the magnesium alloys [52]. In the present study, the potentiodynamic polarization curves were firstly measured to investigate the corrosion behaviors of the different magnesium alloys. In general, a higher corrosion potential ($E_{corr}$) often means a more stable thermodynamic state, while a lower corrosion current density ($i_{corr}$) means that the sample may corrode more slowly. Figure 5 shows the potentiodynamic polarization curves of the magnesium alloys with the different conversion layers, and Table 2 shows the corrosion potentials and corrosion current densities determined by the Tafel extrapolation method. It can be discovered that the corrosion potentials of all the chemical conversion layers were higher than that of the original magnesium alloy (−1.573 V), the corrosion current density ($8.16 \times 10^{-4}$ A·cm$^{-2}$) and the annual corrosion depth (18.5 mm/y) of the original magnesium alloy was also the highest, indicating that the chemical conversion treatment can improve the corrosion resistance of the magnesium alloy to varying degrees. This was due to the fact that the natural passivation layer on the pristine magnesium alloy was lacking density and susceptibility to corrosion in corrosive liquids. Additionally, the thermodynamic stability of the magnesium alloy increased thanks to the surface chemical treatment, which also slowed down the corrosion rate. As compared to the blank magnesium alloy, the corrosion current density of the chemical treated samples generally

decreased by several orders of magnitude, indicating that these conversion coatings had excellent corrosion resistance when immersed in SBF solution. These coatings partially or completely isolate the magnesium substrate from the corrosion medium, preventing the corrosion medium from attacking the magnesium alloy substrate surface and thus lowering the corrosion rate. The $MgF_2$ coating on the Mg-HF surface is often created by the interatomic displacement of Mg and HF, and $MgF_2$ is insoluble in water, therefore it has a good shielding effect to slow down the corrosion rate. Additionally, Yan et al. [53] found that the presence of NaCl, $MgSO_4$, $Na_2HPO_4$ and $NaHCO_3$ in the simulated bodily fluids can effectively prevent the dissolving of $MgF_2$ coatings. The corrosion potential of Mg-P (−1.329 V) was higher than that of Mg-HF (−1.553 V), indicating that the phosphate film has a better thermodynamic stability compared to the $MgF_2$ film. Additionally, the laminar structure coating of the phosphate treatment contributed to increase the surface roughness, which can benefit the interfacial bonding of the magnesium matrix. The corrosion current density of Mg-PA ($4.413 \times 10^{-7}$ A·cm$^{-2}$) was three orders of magnitude lower than that of Mg, which resulted from the fact that phytic acid can chelate magnesium to form a stable phytic acid complex on the surface, and the excellent hydrophilicity was useful to improve the adhesion between the magnesium substrate and the coating, thereby reducing the corrosion rate. Among all samples, Mg-OH had the highest corrosion potential (−1.325 V), its corrosion current density ($7.901 \times 10^{-11}$ A·cm$^{-2}$) decreased by seven orders of magnitude compared to the pristine magnesium, and annual corrosion depth was only $1.79 \times 10^{-6}$ mm/y, indicating that the chemical conversion layer after alkali heat treatment had the best anticorrosion performance. In addition, the alkali heat treatment also had the smooth surface and the smallest water contact angle, which can also contribute to the improvement of the corrosion resistance of the magnesium alloy.

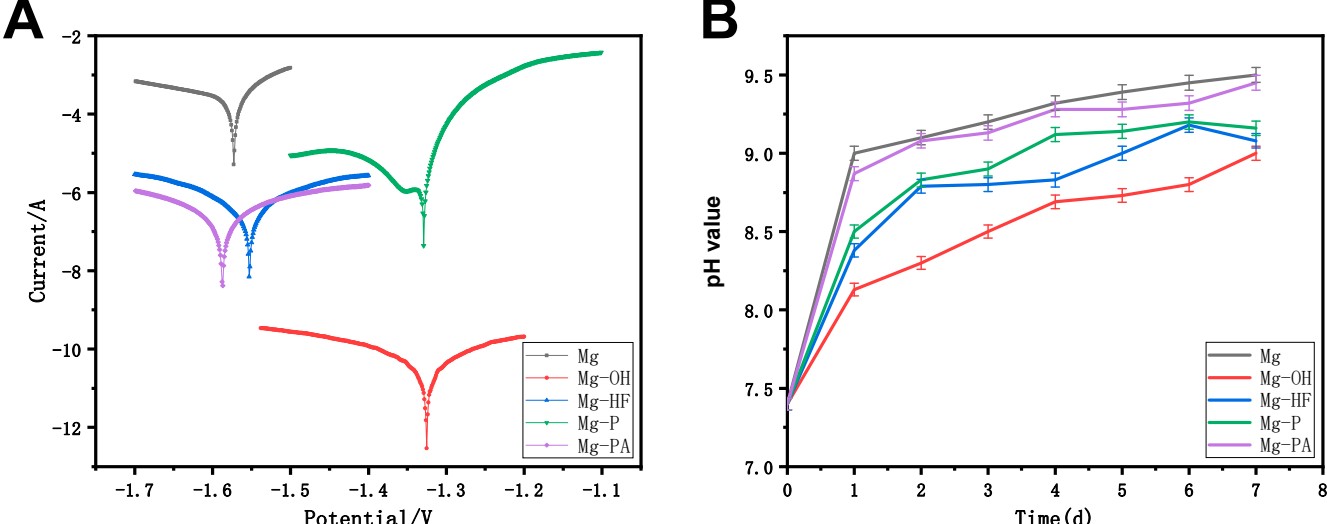

**Figure 5.** (**A**) The potentiodynamic polarization curves of the different samples. (**B**) The pH curves of the different samples immersed in SBF solution for 7 days.

**Table 2.** Corrosion potential and corrosion current density of the different samples.

| Samples | $E_{corr}$/(V) | $i_{corr}$/(A·cm$^{-2}$) | d/(mm/y) |
|---|---|---|---|
| Mg | −1.573 | $8.16 \times 10^{-4}$ | 18.5 |
| Mg-OH | −1.325 | $7.901 \times 10^{-11}$ | $1.79 \times 10^{-6}$ |
| Mg-HF | −1.553 | $3.000 \times 10^{-6}$ | $6.79 \times 10^{-2}$ |
| Mg-P | −1.329 | $1.798 \times 10^{-6}$ | $4.07 \times 10^{-2}$ |
| Mg-PA | −1.587 | $4.413 \times 10^{-7}$ | $9.98 \times 10^{-3}$ |

The physiological environment of the human body has a weak alkaline pH of roughly 7.4. According to the corrosion mechanism of the magnesium alloy, corrosion degradation results in a significant amount of corrosion products, including hydrogen gas and hydroxide, which raises the pH value and thus may cause a series of adverse physiological reactions. Figure 5B displays the pH change curves for the different samples dipped in SBF solution for seven days. It is obvious that the pH value for the magnesium alloys both before and after the chemical treatments increased with the increase in the immersion time. As the surface conversion film started to dissolve at the initial stage of the immersion, the structure may become loose, the corrosion medium permeated the coating, and subsequently corroded the magnesium substrate. The corrosion of the magnesium alloy was often brought on by an oxidation-reduction reaction in which the anode released $Mg^{2+}$ and the cathode precipitated hydrogen [54]. As water gains electrons, gaseous hydrogen and hydroxide ions were produced. The pH value of each sample both exhibited a rising trend due to the formation of $OH^-$, and the corrosion products that were gradually accumulating on the magnesium alloy surface caused the degradation rate decreased generally and the curve rose slowly. The pH value of the pristine magnesium alloy grew the fastest, rising to 9.5 in just 7 days, suggesting that the unmodified magnesium alloy had the lowest anticorrosion performance and the quickest degradation rate. The other modified samples showed a slower pH rising trend, indicating that the surface chemical conversion layer could prevent the corrosion of the magnesium alloy and then reduce the degradation rate to different degrees. From 2 days to 4 days, the rising trend of the curve slowed down because $SO_4^{2-}$ and $HCO_3^{2-}$ in the simulated body fluid can inhibit the corrosion to some extent. Song et al. [55] also found that sulfate or bicarbonate replaced chloride ions adsorbed on the magnesium surface and passivated the active sites on the magnesium surface, resulting in a reduction in the corrosion rate. As the immersion time increased, the corrosion products increasingly accumulated and made it harder for the corrosion medium to contact with the magnesium substrate. This resulted in a slower degradation rate, which was the reason for the stable pH curve of Mg-HF. After 6 days, the dissolving of the soluble $MgHPO_4$ on the magnesium alloy surface and the reformation of the less soluble $Mg(OH)_2$ product can substantially slow down the degradation rate of the coating, which may be the reason of the drop in pH value of Mg-HF and Mg-P. The pH value of the alkali heat-treated samples was consistently lower than other coated magnesium alloys, showing that the Mg-OH samples was the most corrosion-resistant of all the coated samples. Additionally, the SEM microscopic images indicated that there were the significant cracks in the phosphate and phytate treated surfaces, indicating that these conversion layers had weak corrosion resistance. In contrast, the sodium hydroxide-treated sample surface was the densest.

Magnesium alloys treated by the different chemical treatment were immersed into SBF solution for 1, 3, 7 and 14 days in order to further characterize the corrosion resistance. The surface corrosion morphologies and elemental contents of the samples after the corrosion were then observed and analyzed by SEM and EDS, respectively, and the results are shown in Figure 6 and Table 3. After soaking for one day, tiny fractures started to emerge on the original magnesium alloy surface, demonstrating that not only the pristine magnesium alloy had poor capacity to withstand the corrosion but also the naturally occurring oxide layer had the inability to withstand the aggressive ions in the solution. Small micro-cracks that formed on Mg-P and Mg-PA surfaces were believed to be the result of local substrate corrosion brought on by chloride ion penetration during the immersion. The Mg-OH and Mg-HF surfaces were no cracks, it was considered that the dense $Mg(OH)_2$ passivation layer and $MgF_2$ coating can reduce the etching of corrosive ions and thus prevent the substrate from corroding. After three days, Mg, Mg-P, and Mg-PA sample surface fractures grew and deepened, and Mg-HF and Mg-OH sample cracks first appeared, indicating that the modified surface was initially quite protective but gradually loses the protection over time. However, the cracked coating was still attached to the magnesium surface, and it still provided a certain degree of protection. The corrosion conditions for all samples became worse with longer immersion times of 7 days. Shrinkage and dehydration of

corrosion products may have an impact on the appearance of wide cracks [41]. Mg surface corrosion was particularly severe, and the appearance of many corrosion products was brought on by the reaction of corrosion ions ($Cl^-$) with the initial layer of $Mg(OH)_2$ on the surface to produce highly water-soluble $MgCl_2$ and release hydroxyl groups. Additionally, Barajas et al. [56] found that mechanical stresses that might cause cracking can be produced by the slow accumulation and expansion of corrosion products on the magnesium alloy surfaces. On the Mg-OH surface, there were small fractures to be observed, indicating that the rough surface of this sample increased the contact area with the corrosive medium and the protective property of the chemical conversion layer became gradually weakening. In 14 days, the coating protection was decreased further. Corrosion pits appeared on the surface of Mg and Mg-P, and the corrosive medium readily contacted the surfaces of the magnesium matrix, increasing the susceptibility to corrosion and encouraging the accumulation and deterioration of the corrosion products. This was most likely because the magnesium matrix acted as an anode with the second phase as a local micro cathode for internal galvanic coupling corrosion, which speeded up its corrosion [57]. The EDS findings demonstrated that the Mg surface corrosion was the most severe, with an increase in the contents of O and Ca and a decrease in the contents of Mg, therefore the corrosion products were probably calcium-rich compounds. Due to surface corrosion and coating deterioration, which allowed magnesium ions from the solution to enter the coating and adsorb to the surface, the Ca content of Mg-P dropped to zero and the Mg content increased. The Mg-HF and Mg-PA coatings had significant amounts of the corrosion products deposited on their surface, likely as a result of the prolonged immersion time. At the end of the reaction, the corrosion products were deposited on the sample surface. Mg, O, P, and Ca were abundant on the surfaces of Mg-HF and Mg-PA, and hydroxides and phosphates, which were corrosion products, may also be present. However, there were only reticulated cracks on the Mg-OH surface, there was no clear surface corrosion, and the EDS data showed no change in the content of Mg, O and C, demonstrating the superiority of the $Mg(OH)_2$ conversion layer in enhancing the magnesium alloy corrosion resistance.

### 3.3. Protein Adsorption

There are three main proteins in human plasma, including albumin, globulin, and fibrinogen. Albumin is a mild, non-reactive protein which can maintain plasma osmolality and ensure normal transport of lipids and steroid hormones. The albumin layer adsorbed on the surface also inhibits the thrombin produced by contact between implants and blood substances, resulting in an antithrombotic effect [58]. Fibrinogen participates in the coagulation and thrombosis processes, and its interactions with platelets can have negative consequences on blood. The amounts of the adsorbed albumin and fibrinogen are shown in Figure 7. It can be seen that Mg-OH and Mg-P had the most albumin adsorbed on them. Albumin can use conformational changes to gain entropy, which enables more albumin to be adsorbed on the hydrophilic surface and delays thrombus formation [59]. On the other hand, albumin molecules are negatively charged and it can interact with $Mg^{2+}$ to adsorb onto the magnesium surface, which may inhibit magnesium anodic dissolution. It has been reported that the albumin adsorption can act as a good shield to block the reactive sites on the magnesium surface, and thus inhibit the penetration of corrosive ions to some extent, finally leading to the improved corrosion resistance [60]. Since proteins have a greater propensity to connect to hydrophobic surfaces and establish stronger interactions, this may explain why Mg-HF absorbed more albumin Figure 7B shows that Mg-HF adsorbed the most fibrinogen because the hydrophobic surface can interact with the hydrophobic fibrinogen. The existence of a powerful attraction between fibrinogen molecules and hydrophobic surfaces was confirmed by Paul et al. [61]. As compared to Mg, both Mg-P and Mg-PA absorbed significantly more albumin and fibrinogen, demonstrating that the modified surface can simultaneously enhance albumin and fibrinogen adsorption. It was noteworthy that Mg-OH had the lowest fibrinogen uptake capacity, along with the

improved albumin adsorption, suggesting that the NaOH-treated magnesium alloy had the ability to selectively adsorb albumin and may therefore have better hemocompatibility.

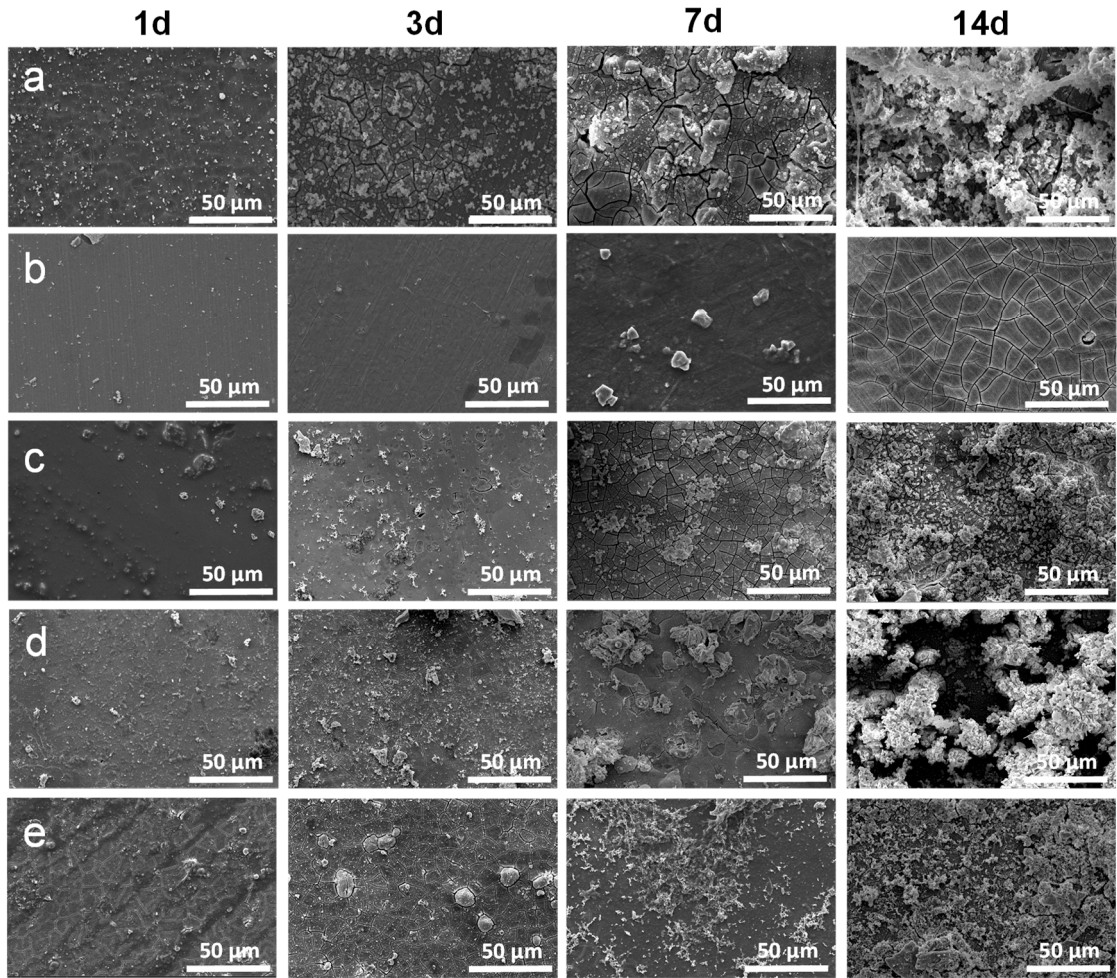

**Figure 6.** Typical SEM images of the different samples immersed in SBF solution for 1, 3, 7 and 14 days: (**a**) Mg; (**b**) Mg-OH; (**c**) Mg-HF; (**d**) Mg-P; (**e**) Mg-PA.

**Table 3.** Surface element contents of the different samples after 14 d immersion in SBF.

| Samples | Mg | C | O | P | F | Ca |
|:---:|:---:|:---:|:---:|:---:|:---:|:---:|
| Mg | 7.23 | - | 66.12 | - | - | 26.65 |
| Mg-OH | 8.83 | 51.95 | 39.22 | - | - | - |
| Mg-HF | 4.24 | 25.64 | 58.21 | 7.22 | 3.52 | 4.68 |
| Mg-P | 23.28 | 24.68 | 52.04 | - | - | - |
| Mg-PA | 9.98 | 30.87 | 50.06 | 3.82 | - | 5.27 |

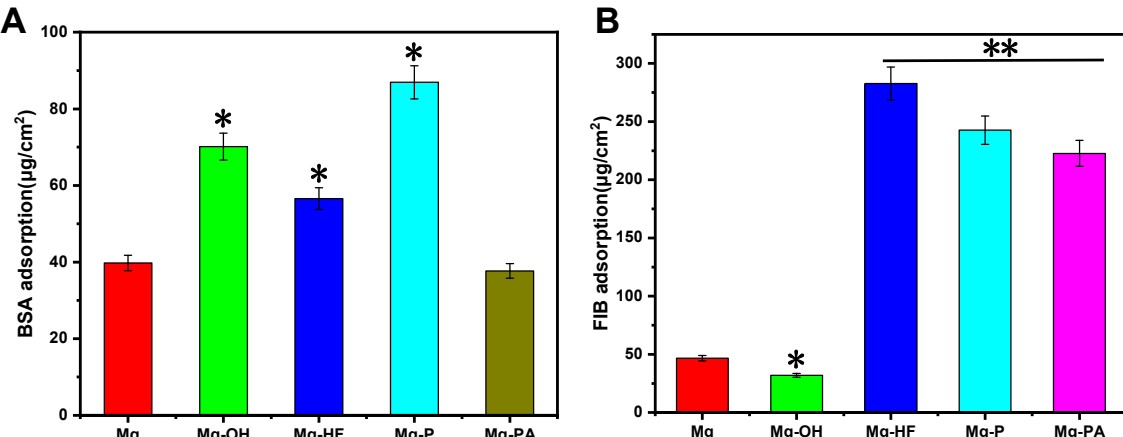

**Figure 7.** The amounts of albumin (**A**) and fibrinogen (**B**) adsorbed on the different surfaces. Data for each sample were taken from three parallel samples, and are expressed as mean ± SD. The samples of Mg-OH, Mg-HF and Mg-P showed significantly differences (* $p < 0.05$) in promoting albumin adsorption compared with the pristine magnesium alloy. The samples of Mg-HF, Mg-P and Mg-PA showed significant differences (** $p < 0.05$) in fibrinogen adsorption compared with the pristine magnesium alloy and Mg-OH.

### 3.4. Blood Compatibility

The implant that comes into contact with the blood must be highly hemocompatible; otherwise, they have the risk of causing a variety of unfavorable reactions, including coagulation, complement activation, platelet activation and leukocyte destruction, which can result in thrombus formation and hemolysis. The hemolysis rate is a crucial factor in determining the hemocompatibility of biomaterials. Better hemocompatibility is indicated by the lower hemolysis. Hemolysis is often influenced by a variety of variables, including chemical compounds, pH levels, and metal ion concentrations, etc. The maximum acceptable hemolysis rate for the blood-contact biomaterials is 5%, in accordance with the ISO 10993-4:2002 standard. Figure 8A shows the results of the hemolysis rates of the different samples. The highest rate of hemolysis can be found on the pristine magnesium alloy, indicating that it may result in severe hemolysis when contacted with human blood. The pristine magnesium can corrode in blood to release a large amount of $Mg^{2+}$, which could result in high osmotic pressure of blood cells and therefore lead to swelling of erythrocytes until membrane rupture, finally leading to the higher hemolysis rate [62]. Additionally, the degradation of the magnesium alloy resulted in the production of $OH^-$, which raised the pH of the blood. The alkaline microenvironment also damages erythrocytes by inducing instability in their membranes [62], which also contributed to severe hemolysis. The surface-modified samples showed a considerable reduction in the rate of hemolysis and all met the hemolysis requirements of the blood-contacting bio-materials, showing that these chemical surface treatments do not seriously harm red blood cells and can effectively prevent the appearance of hemolysis. Due to the better corrosion resistance of $Mg(OH)_2$ layer after NaOH treatment, the decreased release of $Mg^{2+}$ can significantly lower the hemolysis rate when the magnesium matrix came into contact with the blood. The hydrophobic of Mg-HF had water-resistant property, which can slow down the degradation rate of the magnesium alloy and therefore reduce the hemolysis rate; on the other hand, the repulsion to blood resulted in low red cells adhesion. The phosphate coating on Mg-P can prevent the $Cl^-$ adsorption by electrostatic repulsion because of the presence of the negatively charged groups ($PO_4^{3-}$). It can obviously hamper the magnesium matrix from harming the red blood cells and hence preventing the thrombosis [8]. The lowest hemolysis rate was found in the Mg-PA samples, indicating that the phytate macromolecules were chemically bonded to the magnesium ions and other cations to form dense chelates that can prevent direct contact between the magnesium matrix and red blood cells, resulting in the least damage to the red blood cells and good anti-hemolytic properties [63].

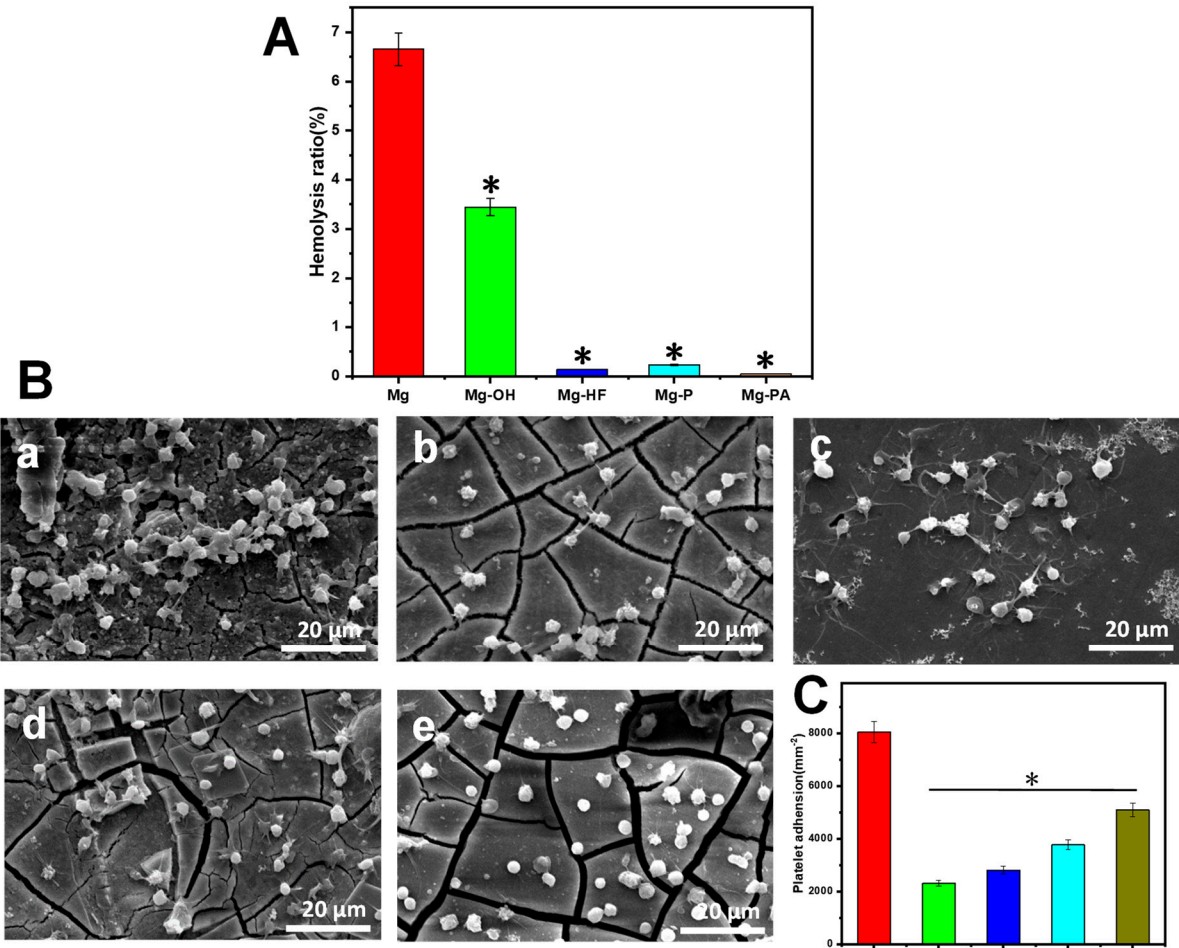

**Figure 8.** (**A**) Hemolysis rates of the different samples. (**B**) Typical SEM images of platelet adhesion on the surface of different samples: (**a**) Mg; (**b**) Mg-OH; (**c**) Mg-HF; (**d**) Mg-P; (**e**) Mg-PA. (**C**) The number of the platelets adhered on the different samples. For hemolysis rate and platelet number, the data are presented as mean $\pm$ SD of three parallel samples. The values of the modified samples are significantly lower (* $p < 0.05$) than that of the pristine magnesium alloy.

Platelet adhesion and activation play a vital role in coagulation responses. When a poorly blood-compatible biomaterial is exposed to blood, platelets can attach to it and undergo shape changes, aggregation, and activation, leading to thrombus formation [64]. The quantity of platelets attached to the surface, morphological changes, and the expression of particular proteins on the platelet membrane can all be used to investigate the degree of the platelet behaviors [65]. In the present study, scanning electron microscopy (SEM) images were taken to reveal the quantity and the morphological changes in the attached platelets on the different surfaces, and the results are shown in Figure 8B,C. It can be clearly seen that the highest number of platelets aggregated on the pristine magnesium alloy surface. It was considered that the most severe surface corrosion and surface hydrophobicity may promote platelet adhesion and activation, and the high pH value caused by an excessively rapid corrosion rate also promoted the platelet activation. This suggested that the pure magnesium alloy had poor blood compatibility, by contrast, there was an obvious reduction in the number of platelets adhered on the chemical treated magnesium alloy surfaces (Figure 8C). The Mg-OH displayed the lowest number of the platelets attached on the surface, and this was because that the alkali heat treatment had created a chemical passivation layer with active hydroxyl groups on the surface, which can improve the hydrophilicity and corrosion resistance and further inhibit platelet adhesion and activation. In addition, it can be seen that almost all of the platelets on Mg-OH were still present and showed no inclination

to become activated, which could be conducive to prevent thrombus formation after the implantation. The Mg-HF and Mg-P samples had comparatively less platelets adhesion and activation as compared to the blank magnesium alloy, indicating that these two chemical treatments can enhance the anticoagulation of the magnesium alloy to some degree. It was concluded that the improved corrosion resistance can significantly prevent the release of $Mg^{2+}$ and local alkalization, which contributed to enhance the anticoagulation. In addition, the magnesium alloy surface was negatively charged after the phosphate treatment, which can also prevent the adhesion of the negative-charged platelets [26]. For the Mg-P samples, phosphate treatment can result in low platelet adhesion and diffusion to create a good blood contact surface and avoid excessive platelet aggregation. Additionally, the generation of hydrogen and corrosion products has a hindering impact on the approaching platelets [36]. In addition, the sample surface showed a small and thin pseudopod on Mg-HF and Mg-P, indicating a slight activation. Since fibrinogen can be attracted to the surface of Mg-PA, which can promote thrombogenic platelet adhesion and aggregation, as evidenced by the increased platelet count and partial activation with pseudopod morphology.

*3.5. Endothelial Cell Behaviors*

3.5.1. Cell Adhesion

One of the most advantageous characteristics of the vascular implants is their propensity to promote the growth of endothelial cells (ECs) on the surface since ECs have natural biocompatibility and constitute the inner wall of the artery. Biomaterials' biocompatibility is demonstrated by the adherence and growth of cells on them [66]. Typically, when cells come into touch with biomaterials, they could alter their shape in order to integrate the materials and cells. Zhang et al. [67] reported the cell adhesion on the surface of magnesium alloy cultured for 6 and 24 h and found that the number of adherent cells was lower at 6 h, while adhesion would be better at 24 h. Therefore, we began to record the cell adhesion from 24 h. The fluorescence staining images of the endothelial cell adhered on the different surfaces after 1 and 2 days of incubation are shown in Figure 9. It can be seen that the cells can attach and grow on all samples, but the original magnesium alloy surface mainly had the rounded endothelial cells because of its limited corrosion resistance. Large amounts of the hydroxide might cause partial alkalinization of the implant, and the formation of hydrogen gas bubbles on the surface of the magnesium matrix may prevent cell adhesion and growth [68]. After 2 days, the number of cells dramatically increased for the modified samples and covered the majority of the surface. It was considered that the surface coating can slow down the degradation rate and increase the number of cell adhesion. On the other hand, a good surface wettability can create a favorable weak alkaline milieu for cell growth and boost cell adhesion and proliferation, therefore, the number of cells on the Mg-OH surface grew better. In addition, the moderately wetted surface induces the adsorption and activation of proteins, especially mucilage proteins, which promotes cell adhesion and improves surface bioactivity [69]. The $MgF_2$ coating on Mg-HF had the capacity to prevent the excessive release of $Mg^{2+}$ and can enhance cell adherence and proliferation on the surface by a good survival interface [70]. In general, the hydrophilic surface facilitates cell adhesion, migration, and proliferation. Cell adhesion and growth for Mg-P and Mg-PA were positively influenced by the hydrophilicity, coupled with the presence of hydrophilic phosphate groups, it can improve the cell adhesion and differentiation. However, if the immersion period was too long. It may corrode readily and have an adverse impact on cell proliferation. When compared to the pure magnesium, all the modified samples displayed a considerable increase in cell density, proving that these coatings can promote endothelial cell adhesion and growth.

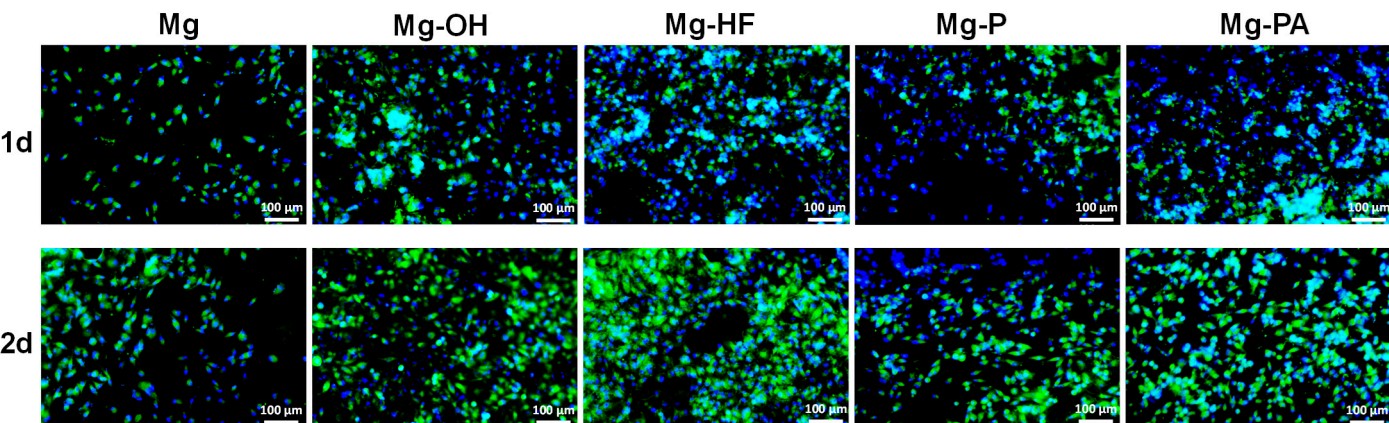

**Figure 9.** The typical fluorescence images of cells attached to the surface of different samples cultured for 1d and 2d.

### 3.5.2. Cell Proliferation

CCK-8 is typically used to the evaluate endothelial cell survival and proliferation. Figure 10 displays the CCK-8 values of the endothelial cells at 1 and 2 days of growth with the different modified magnesium alloys. It was obvious that the pristine magnesium had the lowest CCK-8 value. On the one hand, it was possible that the original magnesium alloy surface corrosion released a large amount of metal ions and hydroxides that were harmful to cell growth. On the other hand, the modification of the magnesium alloy surface can create a chemical conversion layer that was resistant to corrosion, which can achieve the goal of reducing the corrosion rate and reducing the release of metal ions, effectively promoting cell proliferation. The highest CCK-8 value can be detected for Mg-OH in 1 day, suggesting that the alkali heat treatment not only introduced the hydrophilic groups to the magnesium alloy surface, but can also promote cell growth and proliferation. Compared to Mg-P and Mg-PA, Mg-HF had a greater CCK-8 value because of the hydrophobic surface and roughness of the $MgF_2$ layer created by the fluorination treatment, which may increase the attraction to sticky proteins and aid in the stimulation of cell adhesion and proliferation. At 2 days, Mg, Mg-P, and Mg-PA had increased cell proliferation, whereas Mg-OH and Mg-HF had decreased cell proliferation. Nevertheless, Mg-OH has the highest CCK-8 value. It was possible that the force necessary for cell adhesion decreased linearly with the increase in the surface wettability, and once the critical shear stress was reached, the cells were unable to adhere to the surface, impairing endothelial cells ability to grow and proliferate normally. This could explain why the proliferation capacity of the Mg-OH sample decreased [71]. Regarding the Mg-HF sample, it was possible that the immersion time was too long, leading to the corrosion of the $MgF_2$ layer, which was not positive for cell survival. Alternatively, the cells may fall off from the surface due to the production of hydrogen and corrosion products, making it difficult for the cells to attach and proliferate [72]. Generally speaking, cells tend to prefer hydrophilic surfaces for adhesion, spreading, and growth. Since Mg-P and Mg-PA surfaces were rougher than Mg-HF, there was more surface area available for cell adhesion and proliferation, leading to higher CCK-8 values than Mg-HF. According to the results of cell adhesion and proliferation, the Mg-OH surface had good bioactivity and biocompatibility, which created an ideal environment for cell adhesion and proliferation, suggesting that the NaOH-treated samples had good properties of promoting endothelial cell adhesion and growth.

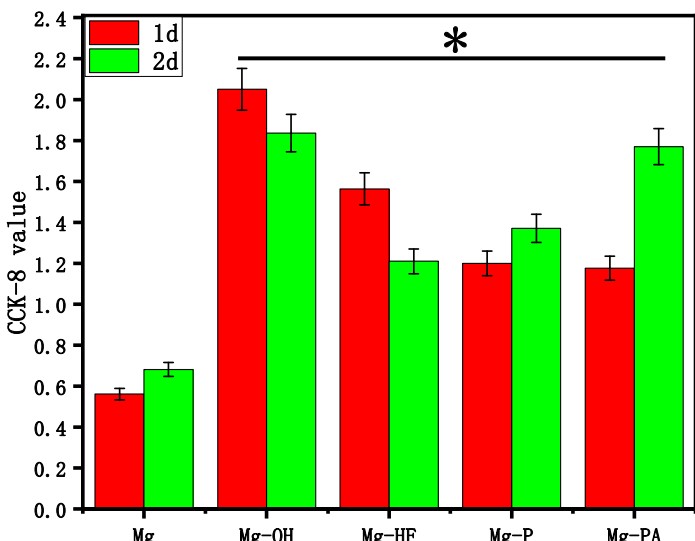

**Figure 10.** CCK-8 values of the endothelial cells cultured on different sample surfaces for 1d and 2d. Data were obtained from three parallel samples and are expressed as mean $\pm$ SD. * $p < 0.05$ indicated the statistical differences compared with Mg.

## 4. Conclusions

The different chemical conversion layers were successfully produced on the magnesium alloy surface by sodium hydroxide (NaOH), hydrofluoric acid (HF), phosphoric acid ($H_3PO_4$) and phytic acid ($C_6H_{18}O_{24}P_6$) treatment, and the as-prepared chemical conversion layers showed the different surface chemical structures and surface morphologies, leading to significant differences in their wettability. The magnesium alloy treated by sodium hydroxide had the best hydrophilicity and a dense structure. Based on the chemical structures of the conversion layers, the different conversion layers exhibited various behaviors of electrochemical corrosion degradation and protein adsorption. In contrast, the magnesium alloy after sodium hydroxide treatment exhibited the best corrosion resistance, the least degree of corrosive degradation, and the capacity to preferentially absorb albumin, which also led to superior anticoagulant characteristics and the ability to withstand corrosion. Although all of the chemically treated samples could enhance the endothelial cell adhesion and proliferation to some degree, the good endothelial cell adhesion and proliferation of the sodium hydroxide-treated magnesium alloy were demonstrated by the comparatively high number of cell adhesions and the highest CCK-8 values. Therefore, among all of the chemical conversion treatments, the surface sodium hydroxide treatment produced a dense and excellent hydrophilic coating that exhibited excellent overall performance and can be applied to magnesium alloy vascular stent materials to improve their biocompatibility and corrosion resistance. In addition, sodium hydroxide treatment can also introduce a large number of hydroxyl groups on the magnesium alloy surface, which makes it easier to further modify the surface by biofunctionalization and thus further enhance the corrosion resistance and biocompatibility of the magnesium alloy.

**Author Contributions:** Conceptualization, L.M. and C.P.; methodology, L.M.; software, L.M., L.L. and F.G.; validation, F.G.; formal analysis, L.M. and L.L.; investigation, L.M. and Q.H.; resources, Y.C. and C.P.; data curation, X.L.; writing—original draft preparation, L.M. and X.L.; writing—review and editing, J.C., Q.Z. and C.P.; visualization, Q.H. and J.C.; supervision, Y.C. and C.P.; funding acquisition, C.P. All authors have read and agreed to the published version of the manuscript.

**Funding:** This research was funded by the National Natural Science Foundation of China (31870952), Natural Science Foundation of Jiangsu Province of China (BK20181480) and Natural Science Foundation of Huaiyin Institute of Technology(22HGZ003).

**Data Availability Statement:** Not applicable.

**Conflicts of Interest:** The authors declare no conflict of interest.

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
