# Peer review of "Comparative Investigation of the Corrosion Behavior and Biocompatibility of the Different Chemical Conversion Coatings on the Magnesium Alloy Surfaces"

_metals, doi:10.3390/met12101644_

Round 1

Reviewer 1 Report

The author has performed good work

This materials are widely used in the medical field, few minor comments to be addressed

1. As per the author claims the contact angle were much higher for the material, i am woundering how come the cell adhesion were started on 1st day itself, author can add more data to support their data in the discussion. 

2. Biocompatibility data on the magnesium alloy surfaces was bit surprising compared within the material, author claims Mg-OH is an fest material, while the cell adhesion figure 9, shows compared with in the material mg-OH shows less cell attachment. author has to explain it why.

3. whether author has performed any leachable study for the materials, if the author claims this can be used in clinical studies in the future, authr has to perform leachable studies with the materials. 

Author Response

Response 1: We gratefully appreciate for your valuable comment. According to our previous study, When the cell culture time is short, the cell adhesion is not firm enough and the number of adherent cells is small. After one day of culture, the cells adhesion was better. Therefore, we began to record the cell adhesion from the first day.

Response 2: That is a very good and reasonable question. In general, fluorescent images can be better used to study the number and morphology of cell adhesion. According to the results of Fig. 9, although the number of cells on the Mg-OH surface seems to be less than those of other chemically treated samples, the cells grow better on the Mg-OH surface based on the CCK-8 results. Therefore, we believe that the cell compatibility of Mg-OH is the best among all materials because CCK-8 can better reflect cell growth.

Response 3: Your comment is valid. The leachable study is a very important and effective method to evaluate the toxicity of biomaterials. In this study, we pay more attention to the direct interaction between materials and cells. On the other hand, due to COVID-19, we are unable to carry out more biochemical experiments in the short term, but we will supplement this experiment in our further research.

Reviewer 2 Report

This is an interesting and rigorous study into the Mg alloy bioresosbable stent implants. The general problem of Mg is high corrosion rate in the human body; it is addressed here by simple chemical conversion coatings.

The following issues should be resolved before the paper can be accepted:

1. The reseachers use AZ31B which contains 3% of Al which is hamful fo the human body. The authors must justify the application of this alloy in the Intoduction.

2. In the abstract, "...sodium hydroxide treated as magnesium alloy..." -> "as" is not needed, please correct the typo.

3. In the Sample preparation, Phosphating and Phytic acid teatment have a preparation step of NaOH surface cleaning. Why do they have 15 and 30 min durations? Should they have the same time? Please comment.

4. Fig. 1 and 2 - XPS analysis should show Al and Zn peaks for the substrate.

5. Table 2 - Corrosion tests should also include the corrosion rate (mm/yr) for comparison with other papers.

6. Biological tests (Fig. 7, 8, 10) must show significant differences (p<0.05), marked, for example, with stars(*).

7. The tolerance intervals in all the figures are shown with too thin lines, very hard to see, please correct.

Author Response

Response 1: Thank you so much for your careful check. We have supplement some information to justify the application of this alloy as follows.

“AZ31B is a kind of widely-used magnesium alloy for developing the cardiovascular implants. When the content of Al element in AZ31B magnesium alloy was about 3%, it could form sosoloid with magnesium, meanwhile Al could increase the strength and plasticity of magnesium alloy and improve the structure stability of the oxide film. Although Al is harmful to human body, the slow release of Al3+ after implantation will not cause great harm to the human body.”

Response 2: Thank you very much for finding this error. We are sorry for this spelling problem and have corrected it according to your comment.

Response 3: Thank you very much for finding this error. The time of the preparation step should be the same (30min), it is my mistake to write. Sodium hydroxide cleaning of polished magnesium alloy, one is to remove surface impurities, the second is to play a part of passivation, conducive to further surface modification.

Response 4: We appreciate your valuable comments. Magnesium alloys have active chemical properties. In the natural environment, the surface will form an oxide film, and at the same time, the surface will also be polluted by hydrocarbons. Therefore, the surface aluminum and Zn contents is very small, resulting in almost no aluminum and Zn elements can be detected on the XPS survey spectra.

Response 5: We gratefully appreciate for your valuable comment. Fitting corrosion currents by Tafel extrapolation and using corrosion current density to calculate the annual corrosion depth. The paper has been modified.

Response 6: Thank you very much for your careful examination. The statistical analysis in section 2.7 was added to the experimental section and the significant differences were modified accordingly. 2.7 "All the data are expressed as mean ± standard deviation (SD) and statistically analyzed using SPSS 12.0. Statistically significant differences were determined by one-way analysis of variance (ANOVA), and P < 0.05 was considered to be statistically significant."

Response 7: Thank you so much for your careful check. The lines of each picture are thickened. The spacing has also been increased for aesthetic purposes.

Reviewer 3 Report

The manuscript "Comparative investigation of the corrosion behavior and biocompatibility of the different chemically conversion coatings on the magnesium alloy surfaces" is devoted to thorough analysis of chemical and biological behavior of chemically treated Mg samples. In particular, corrosion, protein and cell precipitation, hemolysis and cell proliferation were studied for bare magnesium as well as magnesium treated with sodium hydroxide, hydrofluoric, phosphoric and phytic acids. Provided data and analysis are of high interest, manuscript is ready for publishing after fixing some minor issues. In particular:

1. Phrase “... which can be attributed to the presence of P-OH group in Mg3(PO4)2 and the hydroxyl groups in Mg(OH)2 layer, which reduced surface roughness and decreased the water contact angle by forming a ligand bond (P-O-P).”

Generally the higher surface roughness of hydrophilic substrate the more hydrophilic it is, so this explanation should be revisited.

2. Phrase “Additionally, the laminar structure coating of the phosphate treatment contributed to increase the surface roughness, which can benefit the interfacial bonding of the magnesium matrix

and increase the corrosion resistance in the physiological environment.”

This correlation of higher roughness (with cracks!) with increase of corrosion resistance is quite dubious. Both roughness and cracks should increase corrosion due to higher effective surface area per apparent surface area and easier access of corrosion media to Mg substrate correspondingly.

3. In figure 6 Mg-OH for 14d image should be checked and elucidated. Where did the cracks go?

4. In figure 8, I recommend to rename 8b to 8c and vice versa.

Author Response

Response 1: Thank you very much for finding this error. We are sorry for this spelling problem and have corrected it according to your suggestion.

Phrase “... which can be attributed to the presence of P-OH group in Mg3(PO4)2 and the hydroxyl groups in Mg(OH)2 layer as well as the increased surface roughness can significantly reduce the water contact angle.”

Response 2: Thank you so much for your careful check. We have made changes to the corresponding parts of the manuscript.

Phrase “…indicating that the phosphate film has a better thermodynamic stability compared to the MgF2 film. Additionally, the laminar structure coating of the phosphate treatment contributed to increase the surface roughness, which can benefit the interfacial bonding of the magnesium matrix.”

Response 3: Thank you very much for finding this error. The 14d corrosion in the SEM picture is cracked and I made a mistake when I put the picture in. We have made the corresponding changes.

Phrase “In 14 days…However, there were only reticulated cracks on the Mg-OH surface, there was no clear surface corrosion…”

Response 4: We gratefully appreciate for your valuable comment. We have corrected it according to your comment and we have handled the layout of the images aesthetically.